# Remote Monitoring of Crop Nitrogen Nutrition to Adjust Crop Models: A Review

**Luís Silva** [1,2,3,4,*] , **Luís Alcino Conceição** [3,4] , **Fernando Cebola Lidon** [1,2] and **Benvindo Maçãs** [2,5]

1    Earth Sciences Department, Faculdade de Ciências e Tecnologia, Campus da Caparica,
     Universidade Nova de Lisboa, 2829-516 Caparica, Portugal
2    GeoBioTec Research Center, Faculdade de Ciências e Tecnologia, Campus da Caparica,
     Universidade Nova de Lisboa, 2829-516 Caparica, Portugal
3    Instituto Politécnico de Portalegre, 7300-110 Portalegre, Portugal
4    VALORIZA—Centro de Investigação para a Valorização dos Recursos Endógenos,
     Instituto Politécnico de Portalegre, 7300-110 Portalegre, Portugal
5    Instituto Nacional de Investigação Agrária e Veterinária, I.P., Estrada Gil Vaz, Ap. 6, 7350-901 Elvas, Portugal
*    Correspondence: lmr.silva@campus.fct.unl.pt

**Abstract:** Nitrogen use efficiency (NUE) is a central issue to address regarding the nitrogen (N) uptake by crops, and can be improved by applying the correct dose of fertilizers at specific points in the fields according to the plants status. The N nutrition index (NNI) was developed to diagnose plant N status. However, its determination requires destructive, time-consuming measurements of plant N content (PNC) and plant dry matter (PDM). To overcome logistical and economic problems, it is necessary to assesses crop NNI rapidly and non-destructively. According to the literature which we reviewed, it, as well as PNC and PDM, can be estimated using vegetation indices obtained from remote sensing. While sensory techniques are useful for measuring PNC, crop growth models estimate crop N requirements. Research has indicated that the accuracy of the estimate is increased through the integration of remote sensing data to periodically update the model, considering the spatial variability in the plot. However, this combination of data presents some difficulties. On one hand, at the level of remote sensing is the identification of the most appropriate sensor for each situation, and on the other hand, at the level of crop growth models is the estimation of the needs of crops in the interest stages of growth. The methods used to couple remote sensing data with the needs of crops estimated by crop growth models must be very well calibrated, especially for the crop parameters and for the environment around this crop. Therefore, this paper reviews currently available information from Google Scholar and ScienceDirect to identify studies relevant to crops N nutrition status, to assess crop NNI through non-destructive methods, and to integrate the remote sensing data on crop models from which the cited articles were selected. Finally, we discuss further research on PNC determination via remote sensing and algorithms to help farmers with field application. Although some knowledge about this determination is still necessary, we can define three guidelines to aid in choosing a correct platform.

**Keywords:** conservative agriculture; crop nutrition; nitrogen crop sensor; machine learning; decision support systems

## 1. Introduction

For European farmers, fertilizer costs represent, on average, around 6% of input costs, and can reach up to 12% for arable crop producers. The European Commission (EC) presented a communication in November 2022 which presented actions and guidelines to optimize the use of fertilizers and reduce our dependence on them while maintaining production [1]. In the Mediterranean area, it is necessary to reduce production costs and the pollution caused by agricultural activity; these goals can be achieved by following conservative agriculture practices. Input management should also be improved by applying

precise doses of nitrogen (N) to each specific point of the field [2]. It is necessary to avoid the uncontrolled and injudicious use of fertilizers without a strategy or method, as the use of nutrient site-specific management is essential to improving soil fertility and crop productivity [3]. A good management strategy can include the use of mineral fertilizers combined with organic manures to increase the total nitrogen stock in the soil, guaranteeing increases in production over time and, consequently, higher net revenues. The use of organic fertilizers also makes it possible to improve the environment of the soil rhizosphere, improving the quality of production. The use of organic fertilizer also makes it possible to recycle nutrients and prevent them from causing pollution, thus improving fertilizer use efficiency [4–6]. Pollution can be significant in the soil, water, and air [5].

N use efficiency (NUE) can be improved both in space and in time through precision N management. This type of management refers to the need to accurately estimate the crop's N needs and ensure that the correct dose of fertilizer is provided. Through this relationship between accumulated N and needs, farmers can more easily reduce yield gaps [7–10]. To facilitate the adjustment of N fertilization, the N nutrition index (NNI) was developed [11,12]. To estimate this index, it is necessary to know the plant N content (PNC) and the crop biomass, which requires substantial time, sophisticated laboratory equipment, and associated costs. The time required for sample collection and analyses may disallow timely responses from producers to crop N deficiencies [13]. To overcome these logistical and economic problems, it is necessary to calibrate the NNI using other methods [14].

NNI can be estimated remotely using optical sensing [9,15–20]. This indirect estimate of the NNI comes from an analysis of the relationship between the vegetation indices (VIs) that remotely detect the PNC, as well as the analysis of the spatially distributed values of N concentration in plant dry matter [21,22]. Previous studies have suggested classifying the non-destructive NNI acquisition methods by corresponding platforms [20].

While sensory techniques are useful for measuring PNC, there are also crop growth models that estimate crop N requirements. The accuracy of the estimate is increased by integrating remote sensing data to periodically update the model, considering the spatial variability in the plot [23]. However, this combination of data has some difficulties [23–25]. Crop models (CMs) deal with different crop growth stages and with different N concentrations in plants (highly dependent on initial growth stages) [26–28]. This integration of data into an N recommendation system for different management zones may involve the use of more complex algorithms, such as machine learning (ML) [29].

In this paper, we will address some issues related to the process of assessing crop N nutrition, especially by non-destructive methods, such as leaf-based and ground-canopy sensors, unmanned aerial vehicle (UAV) sensors, and satellite platforms. The remote PNC estimates vary significantly throughout the crop cycle, and may show insufficient values in the initial crop growth stages of development when soil vegetation cover is reduced. These estimative values can also be changed in more advanced states through index saturation. They may also vary according to the measuring devices used and their resolution. We will also focus on remote sensing data integration in CMs, as well as the constraints and uncertainties of this integration. It is necessary to clarify the relationship between the crop N status and the N dose to be applied, depending on the crop growth period, and to achieve an in-season production process. This is the importance of using CMs. However, a problem that arises is the inability of linear methods to join remote sensing data with CMs in order to be able to consider most of the problems that affect crops beyond the soil–plant–atmosphere relationship. There is a lack of knowledge on the use of sensors that provide digital information regarding the parameters of interest to adjust the models. Given the specific conditions of each region and the context in which nitrogen fertilizers are involved, these technologies must be made available to farmers so that they can be used in crop production.

There is a need to identify the state of the art of integrating crop N status with remote sensing data in crop model approaches for the purpose of precision top-dressing

fertilization of Mediterranean crops. To obtain insight, studies were analyzed from several dimensions. Firstly, the following three research questions (RQs) were defined:

- RQ1: Is it possible to measure PNC exclusively by non-destructive methods?
- RQ2: What resources have been used in the recent literature to assess crop N nutrition status?
- RQ3: What are the challenges in the prediction of the Mediterranean crop N nutrition status using only non-destructive methods, and how accurately is it possible to measure this status?

The core of these questions is discussed, and the questions are answered, in Sections 6 and 7.

## 2. Materials and Methods

A literature review was conducted using Google Scholar® and ScienceDirect®, focusing attention on the most recent articles (2017–2023) in which the most recent technological approaches were assessed. The keywords used for the search were "Nitrogen Nutrition Index", "Remote Sensing", "Crop Model", "Nitrogen Use Efficiency", and "Mediterranean". Although several studies were found, the introduction of the term "Mediterranean" in the keywords reduced the number of studies resulting from the analysis more than expected. The application of NNI, remote sensing, CMs, and ML was not only considered for Mediterranean crops, but they were prioritized.

To exclude irrelevant studies, the studies were analyzed and graded based on removal criteria (RC), as follows:

- RC 1—Publication is not related to the sustainability of the agricultural sector;
- RC 2—Publication is not related to crop nitrogen nutrition status;
- RC 3—Publication is not written in English;
- RC 4—Publication is a duplicate;
- RC 5—Full text of the publication is not available.

After applying the RC, the remaining articles were examined in greater depth to identify the ones most representative of each theme. A total of 148 references were selected for further analysis, 32 of which were review articles.

## 3. Estimation of Crop N Nutrition Status

### 3.1. NUE

N is one of the decisive elements of plant growth, crop biomass accumulation, and yield formation. Managing N as an input in crop production requires extensive knowledge of what influences crop N status. Understanding N management in relation to N supply and demand, as well as the underlying processes governing N uptake and distribution in plant organs, is imperative for quantifying the dynamics of N in the cropping system. N fertilization management at key growth stages is a growing practice for increasing NUE. It is a function of the N uptake from natural and fertilizer-borne sources, N assimilation, and plant physiology-specific rates of translocating N into harvestable biomass [20]. The complexity assumed in the calculation of the NUE may vary according to the detail required in the estimated result. Complexity can be increased by adding environmental variables and geochemical measurements [30]. Moreover, localized environmental resources, such as topographic and climatic attributes, were considered to predict habitat suitability [31]. The level of complexity of the algorithms must be in accordance with the needs of the farmers and their knowledge of how to deal with them.

Due to the importance that agricultural ecosystems have for the survival of human beings and the development of the social economy [32], algorithms that take economic, agronomic, and environmental aspects into account are more accurate and reliable [29]. The change in the quality of the environment depends on human attitudes [33]. Conceição et al. [34] indicated that the rainfed characteristics of the Alentejo region cause it to require great care in terms of controlling nitrate concentrations, especially in vulnerable

areas. These characteristics reduce the efficiency of nitrogen top-dressed fertilizations, which suggests new forms of mechanization based on precision farming technologies, namely, variable rate application (VRA). VRA technology detects information about a given crop system and allows a system to make more informed decisions.

As mentioned previously, the most appropriate method to calculate NUE is determined according to the context in which the research is inserted, but also by the available data. Usually, the calculation of this efficiency is strongly related to the N recovery efficiency. In this calculation, the absorption of N is measured in relation to the available quantity of the nutrient during fertilization [30]. Parallel to this measurement, isotopes can be used, which are a useful tool in agricultural studies, as they enable the tracing of a particular element behaving similarly to its non-isotopic analog through various pathways in order to obtain quantitative measurements. The $^{15}$N isotope is stable and occurs naturally, with an abundance of 0.3663% [30]. Mass spectrometry is the most common and precise method to measure this stable isotope. The use of $^{15}$N-labeled fertilizer allows us to distinguish whether recently mineralized N or initial soil-borne N is taken up by the plant [30].

The procedure followed for the calculation of this fraction and other derived parameters for N using $^{15}$N-labeled materials is given below [30]:

- Measurements needed for experiments with $^{15}$N:

1. PDM yield for the whole plant or sub-divided into plant parts;
2. Total N concentration (% N in PDM) of the whole plant or plant parts, as in point 1; this is determined by chemical methods, e.g., Kjeldahl, or by combustion (Dumas);
3. Plant % $^{15}$N abundance, which is analyzed by emission or mass spectrometry;
4. Fertilizer % $^{15}$N abundance;
5. $^{15}$N-labeled fertilizer(s) used and N rate(s) of application.

- Calculations for experiments with $^{15}$N:

6. % $^{15}$N abundance is transformed into atom % $^{15}$N excess by subtracting the natural abundance from the % N abundance of the sample. Afterwards, the following calculation (Equation (1)) can be made:

$$\%Ndff = (atom\%^{15}N_{excessplant}/atom\%^{15}N_{excessfertilizer}) \times 100, \tag{1}$$

where %Ndff is % N derived from the fertilizer, for instance, if Ndff = 0.25, this means that 1/4 of the N in the plant came from the fertilizer. If soil and fertilizer were the only sources of N available to the plant, then the remaining 3/4 of the N in the plant came from the soil. If these fractions are expressed in percentages, then %Ndff = 25% and %Ndfs = 75%, where %Ndfs is % N derived from the soil.

7. PDM yield per unit area (Equation (2)):

$$\text{PDM yield (kg/ha)} = \text{FW (kg)} \times [10{,}000 \ (m^2/ha)/\text{area harvested} \ (m^2)] \times [\text{SDW (kg)/SFW (kg)}], \tag{2}$$

where FW is the fresh weight per area harvested and SDW and SFW are the subsample dry and fresh weight, respectively.

8. N yield per unit area (Equation (3)):

$$\text{N yield (kg/ha)} = \text{PDM yield (kg/ha)} \times [\%N/100] \tag{3}$$

9. Fertilizer N yield per unit area (Equation (4)):

$$\text{Fertilizer N yield (kg/ha)} = \text{N yield (kg/ha)} \times [\%Ndff/100] \tag{4}$$

10. Utilization of N from the fertilizer (Equation (5)):

$$\% \text{ Fertilizer N utilization} = (\text{Fertilizer N yield/Rate of N application}) \times 100 \tag{5}$$

### 3.2. Critical N Concentration and Critical N Dilution Curves

The concept of critical N concentration ($N_c$) is essential to calculate the N state of the crop from the PNC. $N_c$ is the N concentration required to achieve maximum plant growth [35]. Lemaire et al. [36] proposed the $\%N_c$ as a negative power function called a "dilution curve" (Equation (6)):

$$\%N_c = a \cdot PDM^{-b}, \tag{6}$$

where PDM is the aboveground plant dry biomass (ton ha$^{-1}$), and $\%N_c$ is the PNC articulated as a percentage of PDM, where $a$ represents the N concentration in the PDM when PDM = 1 ton ha$^{-1}$, and $b$ is a statistical parameter that influences the slope of the relationship [37]. The soil water status might have affected this slope [38]. The inter- and intra-annual variability of precipitation patterns and the frequent occurrence of heat stress periods are characteristics of the Mediterranean climate and promote water deficit situations with significant productivity reductions. This unfavorable distribution tends to increase with climate change [39].

Dilution curves for arable crops have been developed by many researchers and compared in different agroecological zones, years, cultivars, agronomic management practices, and climatic conditions, and are significantly positively correlated with curve parameters $a$ and $b$ [20,40]. Moreover, wheat genotypes with high PNC in the early growth period had a higher value of $a$ and $b$ curve parameters [40,41]. Fernandez et al. [41] highlighted that it is important to use data from young stages when estimating the curves. In these phases, $\%N_c$ is high and PDM is quite low. However, observations below 1 ton ha$^{-1}$ should not be incorporated for model fitting.

C3 plants have different parameters for the maximum N dilution curves compared to C4 plants [42]. Lemaire and Gastal [43] proposed parameter values for various C3 ($\%N_c = 4.8PDM^{-0.34}$) and C4 ($\%N_c = 3.6PDM^{-0.34}$) crops. When comparing similar biomasses, the observed difference between these two groups appears as a difference in the N concentration of the metabolic tissues, as well as in the structural compartment proportion in the total plant biomass. The coefficient a varies between C3 and C4 species according to the differences in the metabolic pathways for CO2 assimilation and the associated differences in leaf anatomy. Under nonlimiting N supplies, C4 crops require 75% of the N required by C3 crops for the same biomass production, so C4 crops contain a smaller proportion of metabolic tissues. Differences in leaf anatomy led to an estimate of 25% lower radiation use efficiency for C3 [43]. Curves proposed by Lemaire and Gastal [43] indicate whether the crop's N status is deficient, optimal, or excessive for the purpose of achieving the maximum crop growth, categorizing plants into these three categories.

Even if the plants are well-nourished with N, research has demonstrated that PNC gradually decreases in the leaf–stem ratio with the crop's maturity, resulting in leaves with high N content and stems with low N content [20]. The uncertainty of the $\%N_c$ is dependent on the biomass level [40]. Additionally, the fewer moments of observation and experimental treatments kept the uncertainty at a high level [40]. A minimum of eight to ten experiments are recommended to estimate an accurate and precise dilution curve. A minimum of three sampling times during the season is advisable. It may be more advantageous to increase the number of sampling times than to increase the number of different fertilization doses above four. It is still necessary to adjust the curve parameters and understand the origin of their differences, and new and more rigorous methods for this achievement are required.

### 3.3. NNI

Using $N_c$ curves based on allometry between plant metabolic and structural compartments, the NNI can be calculated as follows (Equation (7)) [20]:

$$NNI = N_a / N_c, \tag{7}$$

where $N_a$ represents the real PNC. This index allows for an accurate assessment of crop N status diagnosis, fertilizer recommendation, and yield [44]. Estimating this index requires the collection of destructive samples, drying and weighing the biomass to grind these samples, and analyzing $N_a$ by the methods of Kjeldahl or Dumas [45]. Although these classic methods have high accuracy for each sample, they only characterize the crop at its collection point, which makes the method very laborious, costly, and time-consuming, and is not practical for evaluating the crop at all specific points in the field [46,47].

An NNI value equal to or close to 1 means that the crop is at the optimum value to achieve maximum yield. This can be used as a yield and quality benchmark to correct crop management gaps [48]. Note that the NNI can only diagnose the crop N status, but is unable to quantify the dose of N to be top-dressed, especially when NNI < 1. It is necessary to clarify the relationship between the crop N status and the N dose to be applied depending on the crop growth period, and to achieve an in-season production process [20].

## 4. Assessing Crop NNI by Non-Destructive Methods

To mitigate the complex and time-consuming procedure of data collection and analysis by classical methods to determine the NNI, we estimated it remotely using optical sensing [6,15–20].

Sensors measure reflected light at certain wavelengths. They are well correlated with the NNI, and are used to calculate VIs [16,19]. Indirect NNI is assessed by the relationship between VIs and PNC values [21,22]. Some alternative strategies to direct field sampling have been successfully used to estimate PNC remotely and non-destructively (Table 1).

**Table 1.** Sensors used in various crops and crop growth stages to estimate indirect NNI.

| Type of Sensor | Crop | Location | Growth Stage | Reference |
|---|---|---|---|---|
| Leaf-based sensor | Wheat | Southwest France | Anthesis | [13] |
| Leaf-based sensors | Wheat | China | Multi-growth stages | [49] |
| Leaf-based sensors | Wheat | Northern Spain | Stem elongation, leaf-flag emergence, and mid-flowering | [16] |
| Ground-level canopy sensors | Perennial ryegrass | Denmark | Multi-growth stages | [19] |
| Ground-level canopy sensors | Wheat | Spain | Tillering | [50] |
| Ground-level canopy sensors | Maize | Northeast China | V5–V10 growth period | [51] |
| Ground-level canopy sensors | Maize | China | V6-V12 growth period | [52] |
| Ground-level canopy sensors | Sweet pepper | Spain | Multi-growth stages | [53] |
| UAV-mounted multi-spectral camera | Red fescue | Denmark | Multi-growth stages | [6] |
| UAV-mounted multi-spectral camera | Perennial Ryegrass | Denmark | Multi-growth stages | [6] |
| Satellite platforms | Rice | Northeast China | Stem-elongation | [21] |

Vegetation reflectance is normally characterized by the absorption of radiation by plant pigments in the visible spectrum (VIS) between 400–700 nm [54], and by high reflectance in the near-infrared region (NIR) between 700 and 1000 nm, which results from the structure of the canopy (mesophyll) and the effect of leaf density [55]. The most important pigments for photosynthesis are chlorophyll *a* and chlorophyll *b*, for which the maximal absorption of chlorophyll *a* in the red region is 662 nm, and in the blue (B) region, 430 nm; for chlorophyll *b*, the values are 642 nm and 453 nm, respectively. However, the relationship between chlorophyll and PNC varies according to the measuring unit, growth stage, and N fertilization level. Most developed spectral indices focus on the so-called "red edge inflation point", where ratios (commonly called indices) between the VIS and NIR regions are calculated [56]. This chlorophyll-based reflectance plays a significant role in crop N status assessment at the leaf scale. However, at the canopy level, some variables act in a way that they do not at the leaf level, such as in terms of leaf area index (LAI), leaf inclination, soil coefficient, and view–illumination geometry. It should be noted that N-related VIs

are highly correlated with N accumulation in the leaf than in other aboveground parts of the plant. These confounding factors can be alleviated by some vegetation indices (VIs). Specifically, in the early growth stages, the exposed soil background accounts for major variance in the signal detected by the sensors. As LAI increases, the signal from the soil background gradually decreases while the multiple scattering caused by the canopy structure increases [57]. Fu et al. [57] indicated five VIs that minimize confounding factors from ground cover (Table 2).

**Table 2.** VIs to minimize confounding factors.

| VI | Abbreviation | Wavelengths |
|---|---|---|
| Ratio Modified Chlorophyll Absorption in Reflectance Index/Optimized Soil-Adjusted Vegetation Index | MCARI/OSAVI | 550, 670, 700, 800 |
| Transformed Chlorophyll Absorption in Reflectance Index | TCARI | 670, 700 |
| Ratio Transformed Chlorophyll Absorption in Reflectance Index/Optimized Soil-Adjusted Vegetation Index | TCARI/OSAVI | 550, 670, 700, 800 |
| Canopy Chlorophyll Content Index | CCCI | 690–730, 780–1400 |

Yu et al. [58] proposed an NNI remote sensing index ($NNI_{RS}$) based on two VIs, which showed high performance in estimating crop N status. However, before making its estimate, this indirect NNI is required to estimate PNC and aboveground dry biomass to directly relate the NNI with the VIs [57].

Previous studies have suggested classifying the non-destructive NNI acquisition methods by corresponding platforms [20,59,60], as will be described in Sections 3.1–3.3. of this document. The main difference between the sensors is in their practical applicability [61].

### 4.1. Leaf-Based Sensors

Leaf-based sensors do not require sampling for laboratory chemical analysis and are faster than traditional NNI measurement methods. Although they do not directly estimate N concentration in leaves, they remotely measure the chlorophyll content in the chloroplasts. Crop reflectance provides information about the chlorophyll content, and, consequently, about the N concentration. However, chlorophyll meter values are easily affected by crop varieties, growth stages, years, leaf thicknesses, leaf positions, and environmental stress [62–64].

Some authors [65–67] have tested the SPAD-502 (Konica Minolta, Tokyo, Japan) and Hydro N-Tester (Minolta, Japan) chlorophyll meters, and have highlighted that SPAD meters tend to be saturated with high chlorophyll content [68]. Other sensors, such as Dualex 4 (Force-A, Orasy, France), are more suitable for measuring the NNI in situations in which high concentrations of chlorophyll are present [69] and when the crops are still at the beginning stages of development [70,71]. Chlorophyll content increases throughout the crops' growing season, so the accuracy of chlorophyll meter estimates tends to increase with foliar N concentration. The accuracy of chlorophyll meters can be affected by factors such as leaf thickness, leaf angle, and leaf texture, which can influence the amount of light reflected. As the crop cycle progresses, although foliar N concentration is high, these factors tend to make measurement by chlorophyll meters more difficult. On the other hand, the accuracy of fluorescence meters can be affected by factors such as the moisture content of the sheet and the presence of other pigments that can interfere with the fluorescence signal. As the vegetative cycle progresses, these factors decrease, facilitating measurement by sensors such as Dualex.

Although Fiorentini et al. [72] found a mean significant relationship between SPAD readings and the chlorophyll concentrations in leaves ($R^2 = 0.47$), it's only accurate to

measure each site-specific at a time [73,74], and only at the one-leaf level [46,47,67,75]. This is a problem for measuring the crop N status accurately and quickly in the entire field.

*4.2. Ground-Level Canopy Sensors*

To overcome the applicability difficulties experienced when using leaf-based sensors, ground-level canopy sensors can be used. These can be active or passive sensors [20].

Passive sensors read the crop reflectance using daylight as the light source, revealing positive results in terms of the ratio between NIR and R. Fitzgerald et al. [76] used FieldSpec Handheld to demonstrate that the canopy chlorophyll content index–canopy nitrogen index (CCCI-CNI) index was able to predict winter wheat canopy N (g m$^{-2}$) from Zadoks 14–37 with an $R^2$ = 0.97. Palka et al. [77] also used a passive hyperspectral FieldSpec HandHeld 2 (ASD Inc., Falls Church, VA, USA), indicating that CCCI-CNI is a promising index for estimating the PNC of winter wheat until Zadoks stage 49. They used the normalized difference vegetation index (NDVI) and normalized difference red edge index (NDRE), which are based on the fundamental principle that the PNC of green leaves is closely related to chlorophyll content. The NIR band-based VIs can effectively detect the nutritional status of durum wheat ($R^2$ = 0.70 on average) [72]. However, the NDVI, NDRE, and CCCI do not account for the physiological dilution of %N as a function of biomass. This makes it difficult to estimate crop PNC throughout the vegetative period because NDVI becomes insensitive to increasing biomass, becoming "saturated". When canopies close, NDVI stops increasing, but PNC continues to increase along with biomass. Remote estimation of biomass using VIs alone is quite difficult, but when combined with crop modeling, very satisfactory results can be achieved [76]. In addition, the efficiency of such sensors is influenced by weather conditions, including cloud cover, dust, and solar zenith angle [78,79]. Due to their high cost, they are mostly used in scientific research rather than on-farm [21].

Active sensors work differently and are independent of natural light [80], reading the light reflected by the crop and emitted by the sensor light emitters, and may require some calibrations [81–83]. A recent literature review [20] identified some active canopy sensors, such as GreenSeeker (Trimble Navigation Limited, Sunnyvale, CA, USA), Crop Circle (Holland Scientific Inc., Lincoln, NE, USA), and RapidSCAN (Holland Scientific, Lincoln, NE, USA), to remotely estimate crop N status and support precision crop management [20].

Readings made by the GreenSeeker sensor allow for the measurement of a crop's reflectance in bands R (650 nm) and NIR (770 nm), and the calculation of two VIs which are highly correlated with winter wheat [67,83]. The use of this sensor is compromised by the noise caused by the water in the background, which occurs during the early growth stages of rice [84,85]. The first index is NDVI, the use of which is compromised in N topdressing, as it can become saturated under conditions of high leaf area index or medium biomass [86,87]. The second index, which is the Ratio Vegetation Index (RVI), has a weak relationship with biomass, especially under conditions of sparse vegetation cover [88].

The choice of the most suitable VIs varies with the growth stage of the crop at the time of the imagery collection, but also with the number of bands included. Sensors with three or more bands performed better than those with two bands [89].

Crop Circle 430 is an example of a sensor with three bands (R (670 nm), Red-Edge (RE) (730 nm)). Some authors [90,91] conclude that VIs based on the RE band of the Crop Circle sensor are more reliable than NDVIs derived from GreenSeeker. Even so, Crop Circle 430 is able to better appraise the N status of winter wheat [83], but is less sensitive to height than Crop Circle 470. Chen et al. [71] suggest combining various sensors, such as Dualex 4 and Crop Circle 430, to better predict the maize NNI rather than using only one sensor. At a high level of accuracy, it is possible to use more complex sensors, such as Crop Circle Phenom, which is capable of reading several parameters, and integrating them with ML methods in management information to improve maize NNI prediction across the different N rates and drainage and tillage practices [92].

Another example of an active sensor is the RapidSCAN, which groups the radiation into 3 bands and is not affected by the 0.3–3 m measuring height [88]. This sensor allowed

for the calculation of the N Sufficiency Index (NSI) through standard VIs, and for the estimation of NNI in rice as well as in wheat [88,93]. The results achieved by Aranguren et al. [16] revealed that when using the VIs to estimate the NNI, the exponential models outperformed the linear models during the entire growth period, and NNI predictions could be made using a single model with absolute NDVI. To improve the accuracy of the NNI estimation in different conditions, such as under diverse soil, weather, and management conditions, the data from this sensor can be integrated with Genotype × Environment × Management information [20].

For real-time measurement of plant N status during N fertilization, on-the-go crop sensors can be installed to measure R and NIR reflectance. The most well-known systems include Yara N-sensor, Crop Circle, Trimble GreenSeeker [94], and, recently, Fritzmeier ISARIA [95]. This last sensor combines on-the-go spectral measurement of the crop stand with soil productivity maps (map overlay mode). As shown by Pedersen et al. [96], the combination of soil information with the diagnosis of plant N status by spectral measurement has brought about the greatest economic benefits of variable rate application of N fertilizers.

### 4.3. UAVs

The use of UAVs is becoming more and more conventional, mainly because it allows for the collection of imagery with high spatial resolution, relatively low operating cost, and near real-time image acquisition, thus providing a solution to the problems faced by ground canopy-based and leaf-based sensors. Types of UAV airframes are numerous, but the two most common are the roto copter and the fixed wing. With each having its strengths and weaknesses, the flight characteristics of roto copters make them more suitable for agricultural research than fixed wings. These tend to be useful for applicate products [97]. When using these instruments to calculate the NNI, there is an essential workflow to load UAV-based data into precision farming machinery or to perform statistical analysis for research purposes. It starts with creating the flight plan, collecting imagery during the flight, combining and processing imagery (stitching), and, finally, extracting the resulting data. Some cloud-based computing services have eliminated the need for expensive processing software and hardware [97].

National regulations decide when and where UAVs can be used, including flight height and speed limitations, line-of-sight and night operation settings, and restrictions near airports and people agglomerates. Ensuring safe missions is necessary to prevent other manned aircraft operating in the same airspace, birds of prey, and remote controller connection disruptions from resulting in UAVs escaping [97].

According to the review by Olson and Anderson [97] of the wide variety of remote sensors that can be used in UAVs, the three most predominant types in agricultural applications are color (RGB), spectral, and thermal cameras. They add that spectral sensors are classified as passive sensors in terms of their source of electromagnetic radiation. The RGB cameras capture light in the VIS region, grouping the radiation into three bands (Red (R), Green (G), and (B), producing images that can be incorporated into automated object-based classification methods and other machine learning methods [98] in the treatment of information and in the generation of IVs [99–103]. The high resolution of the captured images is unattainable by other cameras, and allows for the creation of digital models of crop heights and improvement of the accuracy of classification software [97]. These cameras allow for the estimation of leaf color, lodging, and canopy cover. Wang et al. [104] demonstrated the usefulness of these high-resolution images in removing background interferences to estimate the N nutritional status of rice plants at the vegetative phase of early crop stages.

Spectral cameras, in this case, multispectral and hyperspectral cameras, capture VIS, NIR, and shortwave infrared (SWIR) segments of the electromagnetic spectrum [97]. These cameras can be used to estimate indirect parameters related to NNI, such as leaf nitrogen content, leaf area index, leaf chlorophyll content, and plant biomass. Furthermore, it can indirectly measure nutrient deficiency in real time by sensitivity to vegetative discolorations

and photosynthetic pigment content. Despite hyperspectral cameras focusing on the same portions of the electromagnetic spectrum as multispectral cameras, they can classify them into hundreds of narrow bands. Hyperspectral cameras are more expensive and require a greater capacity to store hyperspectral data cubes and perform data processing. However, this type of camera makes it possible to distribute the light reflected by the surface into hundreds of bands, grouping them together to make the reading of agronomic parameters of crops more comprehensive [105]. Cilia et al. [106] used hyperspectral sensors to monitor maize N status, and their results showed that the modified chlorophyll absorption ratio index/modified triangular vegetation index 2 (MCARI/MTVI2) and MTVI2 were related to N concentration ($R^2 = 0.59$) and PDM ($R^2 = 0.80$), respectively. In the Mediterranean area, Fiorentini et al. [72] demonstrated that the modified soil-adjusted vegetation index 2 (MSAVI2) showed a better correlation with the chlorophyll concentration (mg g$^{-1}$) of durum wheat leaves, with an $R^2$ value of 0.68, and for the NNI, the NDRE vegetation index performed better, with an $R^2 = 0.85$.

Pereira et al. [104] assessed the performance of UAV and satellite platforms in predicting the N parameters in pasture fields cultivated under an integrated crop–livestock system, considering PNC, PDM, and NNI. The UAV multispectral data resulted in the best prediction accuracies ($R^2 = 0.84$: PNC, 0.70: PDM, and 0.84: NNI). The combination of UAV_RGB with either PlanetScope ($R^2 = 0.79$: PNC, 0.67: PDM, 0.77: NNI) or Sentinel-2A ($R^2 = 0.76$-PNC, 0.57-PDM, 0.69-NNI) improved the performance of the three platforms individually. The association between high spatial and spectral resolutions contributed to the highest prediction accuracy in estimating N variability in pasture fields using remote sensing data. UAV-based remote sensing and advanced computational algorithms, including artificial intelligence, ML, and deep learning, are progressively being applied to make predictions in many farming industries. UAVs with various advanced sensors, including RGB, multispectral, hyperspectral, and thermal cameras, have been used for crop remote sensing applications [105]. Specifically, research with spectral sensors highlights the red-edge chlorophyll index (RECI), the blue nitrogen index, and the red-edge normalized difference index (RENDVI), as well as their very strong relationship with crop N status for ryegrass (*Lolium Perenne* L.), rice, and sorghum (*Sorghum bicolor* L.). These and other indices have shown strong relationships depending on the crop growth stage, the choice and definitions of indices and parameters, and sensing time [97].

### 4.4. Satellite Platforms

Satellite platforms allow for the remote sensing of crops by free image acquisition, and, consequently, estimate crop N status in large-scale plots. Satellites such as Sentinel-2, launched by the European program Copernicus and the European Space Agency (ESA), provide more information on the detection of cultivation parameters, with a high resolution and a short revisit time [107,108]. Currently, Sentinel-2 has a revisit time of 5 days, capturing 13 spectral bands with 10 m of VIS/NIR bands, 30 m of NIR/SWIR bands, and 60 m of atmospheric bands as the ground spatial resolution [109]. Through these bands, Crema et al. [110] calculated CCCI and NDRE indexes in maize, allowing NNI to be estimated directly from remote sensing ($R^2 = 0.76$ and 0.79, respectively). Meier et al. [111] demonstrated advantages in terms of area coverage by increasing the spatial resolution from the current 10–20 m to the ideal 5 m for all spectral bands in a future upgrade of the Sentinel-2 satellites. For the next generation of this constellation, the EC [112] considered an increase in the spatial resolution of the VIS to SWIR bands to 5–10 m.

Other satellites, such as RapidEye, have a higher spatial resolution (5 m) and a revisit time of only 1 day. Characteristics that make this satellite one of the most suitable for assessing both biomass variation in vegetation [112] and N content detection [113–116] allow it to be used on a large scale [117]. Statistical analysis has indicated that both MCARI and the enhanced vegetation index 2 (EVI2) were the best VIs for estimating PNC and PDM, respectively, in areas typically characterized by a Mediterranean climate. However, published results using VIs from RapidEye imagery to assess vegetation status and crop

parameters are still limited. Given the periodicity of the images and their resolution, in arable crops, for example, it can be very interesting to estimate the NNI using PlanetScope, which has a daily frequency and a spatial resolution of 3 m [118].

In recent years, the determination of crop N has become interesting, and has grown using spectroscopy (hyperspectral) imaging [119]. The ESA is conducting studies for the candidate Copernicus Hyperspectral Imaging Mission (CHIME) [120] and Land Surface Temperature Mission (LSTM) [121]. The ESA has become a world leader in Earth observation, especially for European areas, with the development of cutting-edge technology capable of capturing information from the Earth's surface with great accuracy and resolution.

Some private missions, such as those already mentioned, i.e., RapidEye and PlanetScope, and others, such as WorldView, are recognized by the ESA. As their satellites are closer to the Earth's surface, they can capture images with a higher resolution. However, acquiring these images is expensive compared to those of ESA, which are free of charge.

Apart from the advantages of satellite platforms, atmospheric phenomena such as the occurrence of clouds may affect the images, restricting their usage [17,95]. Even when recently launched satellites surpass these obstacles, the difficult situation of temporal resolution becomes worse when these phenomena occur during critical stages of crop development. Without the collection of images at critical growth stages, an effective management strategy based on satellite data becomes impossible. In addition, the presence of surface water in the field, such as the surface water layers in rice crops [20], influences plant reflectance.

In summary, each type of NNI indirect acquisition platform has its applicability. Each is adapted to different crop monitoring needs and maintain great precision in the indirect estimation of the crops' N status, as shown in Table 3.

**Table 3.** Applicability of different indirect acquisition platforms for crop N status.

| Sensor | Parameter Remotely Detected | Index | $R^2$ | Spatial Resolution | Reference |
|---|---|---|---|---|---|
| Leaf-based | Leaf chlorophyll content | - | 0.47 | Leaf-level | [72] |
| Ground-canopy | NNI | CCCI-CNI | 0.97 | cm | [76] |
| Ground-canopy | NNI | NDVI; NDRE | 0.70 | cm | [72] |
| UAV | PNC | MCARI/MTVI2 | 0.59 | cm | [106] |
| UAV | PDM | MTVI2 | 0.80 | cm | [106] |
| UAV | Leaf chlorophyll content | MSAVI2 | 0.68 | cm | [72] |
| UAV | NNI | NDRE | 0.85 | cm | [72] |
| UAV | NNI | | 0.84 | cm | [104] |
| Satellite | NNI | CCCI | 0.76 | 10 m | [110] |
| Satellite | NNI | NDRE | 0.79 | 10 m | [110] |

## 5. Integrating Remote Sensing Data in CMs

### 5.1. Processing Remote Sensing Data

Remote sensing technologies manage to generate a significant amount of data that aids in making well-informed decisions, but diminishes the ability to integrate and analyze the growing amount of data coming from different sources. A combination of this information can be used to make N fertilization recommendations, through algorithms [122,123] highlighting the importance of data analysis methods by ML [124,125]. New advances in ML have increased the possibility of estimating agronomic parameters from digital information. It is necessary to continue research on digital variables from direct measurements and to select the one that is most relevant for each decision. Researchers should focus on estimating agronomic parameters of interest using sensors to inform decision systems, and these should be implemented and adapted to specific points [126].

ML can unravel non-linear problems in diverse datasets [127]. Random forest (RF), support vector machine, and artificial neural network regression have been employed

for the estimation of N status and NNI. Research has demonstrated that the least squares support vector machine may be a promising tool for quantifying N status and evaluating NNI values [127–129]. Ge et al. [130] demonstrated reliable performance of the RF model in estimating PNC by merging the VIs with the R, Green, and B bands. Using the RF model to relate $N_c$ estimated by UAV imaging with measured $N_c$, they obtained coefficients of determination of 0.84 and 0.82 in the first and second years of study, respectively. Compared with other linear or ML regression models, some studies have shown that RF is able to better estimate the crop N nutritional status [131,132]. However, estimating $N_c$ in the most advanced phenological stages results in values below the measured $N_c$. Ge et al. [130] presented two possible reasons: (1) the problem of VIs saturation and (2) major errors and uncertainties in the RF model caused by changes in the structural plant's morphology (such as leaf senescence or panicle or ear emergence).

It is necessary that research reflects field conditions, which has not been the case in trials conducted in small plots [94]. N plot experiments are very useful for studying the influence of different doses of N on plant growth, facilitating the collection of destructive samples, and remote sensing. Although studies performed on small plots using nearby sensors usually allow for more accurate results than the results obtained in studies using remote sensing on farmer fields, it is necessary to determine the modeling accuracy on a larger scale in farmer fields [15].

In this sense, some research projects have emerged recently, such as the MechSmart Forages Project [133], which identifies the need to review traditional cultural itineraries in forage production with new technological-based proposals based on soil, crop, and soil monitoring for the rational use of production factors. This need is amplified by the context of climate change and by the need to sustainably intensify the production of these crops, which support the extensive animal production systems in the Alentejo region. Following this project, another was put into practice, ISOmap Forragem [134], which reinforced and consolidated the knowledge on the integrated use of precision agriculture, mechanization, and digitization technologies in technical itineraries of conservation agriculture during the production of forage. The main objective of the ISOmap Forragem project was to design methods for approaching and using soil and crop monitoring technologies, and for the application of variable dose factors that guarantee the productivity and sustainability of crops in Mediterranean regions. These projects are of great importance in studying the methodology for applying variable rate factor application technologies and transferring this knowledge to the end user, namely, the farmer who needs to make decisions about the quantities of production factors to apply. Given the specific conditions of each region and the context in which nitrogen fertilizers are involved, these technologies must be made available to the farmer so that they can be used in crop production.

The diagnosis of crop N status in farmer fields leads to the guiding of in-season top-dressing N applications through NNI maps that identify different management zones with different nutritional statuses. However, these maps only indicate the status of the crop, and not the amount it needs to reach optimal nutritional status [15]. An N recommendation system sensor–approach fusion mainly consists of using the proposed algorithm differently for different management zones, as defined by soil variability. It may involve the use of more complex combinations, such as CMs, ML, and crop monitoring, to improve the large-scale crop N diagnosis models [15,29,125,135–137].

## 5.2. Modeling Plant Nutrition and Requirements

While sensory techniques are useful for measuring $N_c$, CMs are useful for estimating crop N requirements [138]. Modeling is the process of developing a mathematical representation of a system. The fact that N is so dynamic in soil–plant systems makes the accurate assessment of crop N status a rather complex task. In this sense, CMs can simulate the dynamic physiological process of crop growth and development based on the quantitative relationships between crop growth and environmental conditions, including climate, soil conditions, information on crop genotype, and management strategies [57].

The International Atomic Energy Agency (IAEA) [30] highlights two main objectives for crop and agricultural system models: (i) to better understand the cause–effect relationships in a system and to provide improved qualitative and quantitative interpretations of the behavior of that system—the result of this type of effort is an increase in knowledge; thus, this is a research-oriented goal; (ii) to better predict system behavior to immediately improve the control or management of the system. The result could be a tool/system (software/hardware product) designed for a specific application [30]. The process of modeling can be divided into four broad stages: studying/building, calibration, testing, and application, in this order. Often, problems encountered at the calibration and testing stages cause models to be returned to the building stage to be corrected, and the whole process needs to be repeated [30].

The algorithms represented in the models express the connection between plant processes such as partitioning, biomass growth, respiration, plant water consumption, and photosynthesis, and environmental variables such as daily temperature, photoperiod, water, and available N in the soil [139,140]. CMs have been developed for several purposes, such as analysis of yield gaps, to support decision making and to reduce the time and costs spent on field research [138]. These models can provide dynamic simulations, as well as predictions of crop canopy N status, in imaginary situations [57].

Many mechanistic and ML methods have been developed because they represent the interaction between soil, plants, and climate in a sophisticated way, but also because computational capacity has increased, which is fundamental to dealing with large data sets. However, it is necessary to ensure the applicability of these models [29].

The availability of correct models adjusted to each specific situation affects the application of simulation models in the cases of farmers. In addition, the correct application of these models also depends on the availability and quality of the information that allows for the model to be run [30,141]. STICS is a model that focuses on the water and nitrogenous balance of the soil–crop system. Inputs consider the climate, the soil, and the cropping system [42]. Most of the problems that affect crops are of a multidisciplinary nature, which demonstrates the importance of biophysical and socioeconomic aspects in addition to the soil–plant–atmosphere relationship. It is usually favorable to approach whole systems, since one part affects all others of the system. It is important that the data collection and experimental procedures are specified so that data handling structures and analytical approaches can be defined and developed. In fact, a major problem in CM application is that some essential model input parameters are difficult or impossible to gather, and the crop information provided by the model is limited to discrete points. A combination of CMs and remote sensing is one of the most promising methods for estimating crop N status in the field [57].

FertiliCalc was developed by the University of Córdoba to calculate the fertilization doses of N, phosphorus (P), and potassium (K). It also estimates soil acidification and N losses [142]. The system includes more than 150 crops from which users can choose and add soil data, and the fertilization strategy for P and K. Next, the fertilizers to be applied were selected. Precipitation is considered in order to calculate N leaching. This system is also the basis for several apps for fertilization calculations developed by public institutions in Spain [142]. This and other CMs are simplified descriptions of natural systems, being limited by the lack of information regarding some variables (e.g., production, field, soil, culture, biotic and abiotic factors, cultural itinerary) [23]. In Mediterranean agricultural areas, there is great spatial variability in the physical and chemical properties of the soil, even in the smallest fields [3]. This increased uncertainty in the simulations can reduce the model's overall accuracy. Updating with model variables from more than one CM greatly increases the accuracy in simulations and the understanding of spatial variability [143,144]. A crucial issue of this multi-modal (MM) simulation strategy is the exhaustive calibration process. It usually requires well-trained users to ensure satisfactory results. The development of an automatic calibration process would ensure model standardization [145].

One approach for spatial simulation of crops is based on the integration of geo-graphic information systems and the application of CMs at the plot scale. It is intended to predict the growth, development, and crop production in different environment and agronomic management scenarios at each site-specific point. Firstly, the area in question is divided into a grid of cells, which are considered the site-specific points; then, in each of these individual cells, the model is run [146].

### 5.3. Integration of Remote Sensing into CMs

Although crop models provide reliable simulation performance, it can become complex to ensure their efficiency. Despite spatiotemporal limitations in observation, remote sensing techniques can be another way to increase the accuracy of the models and to periodically update the model, considering the spatial variability in the plot [23,147]. However, this coupling presents some difficulties [23–25]. CMs deal with different crop growth stages and with different PNCs that come from the spectral bands. These measurements are highly dependent on the growth stages, and may show insufficient results in the initial stages when the vegetation cover is very low [26] or be influenced by the phenomenon of saturation in more advanced phenological stages [27,28]. There is still a need for some knowledge regarding PNC determination via remote sensing. Only if these measurements are valid can CMs be adjusted to make them more useful as precision N fertilization prescription techniques [145].

Integration of remote sensing data into crop growth models is feasible through ML models. However, it is important to bear in mind that the result is only reliable if all the data are correct and represent reality. First, the remote sensing data must be collected by the most suitable sensor, which must capture the desired radiation to estimate the desired crop parameter. As for the growth models used, they must consider the characteristics of the environment surrounding the crop and the management practices that are representative of the region in which they are applied, as well as the parameters of the crop in question. For example, it is not sensible to use a CM adapted for covered crops (in greenhouses) in crops grown in the ground in an extensively rainfed situation in the Mediterranean region. Finally, the integration of the remotely detected crop status with the crop needs estimated by the CMs made by ML methods is only possible with well-calibrated models adjusted to each situation. From the review which we have carried out, it can be noted that RF models can constitute a very viable option to integrate measurements of the nutritional N status of crops with their need to achieve optimal production.

Better integration of several types of data, sensors, and algorithms could be carried out to validate field data and to help the large-scale application of algorithms [29]. To integrate the data on recommendation systems, Cortini et al. [29] identified three examples with increasing levels of complexity. Firstly, sensor fusion combines free remote sensing products with low-cost proximal sensors as inputs into an empirical N recommendation system. Secondly, algorithm fusion combines empirical N recommendation systems with ML techniques to contribute knowledge regarding field properties. Finally, high-level data integration combines soil and crop sensing (either proximal or remote) with a crop model.

In January 2020, the EU launched the Farm Sustainability Tool for Nutrients (FaST), aiming to generate fertilization recommendations based on satellite images, crop growth models, and meteorological data. Supported by the European Space Program and the EU ISA Programme, the FaST digital platform will provide resources for agriculture, environment, and sustainability of European farmers, member state paying agencies, agricultural consultants, researchers, and developers of digital solutions. It is intended to be a world-leading platform to generate and reuse solutions for agricultural sustainability and competitiveness based on spatial data (Copernicus and Galileo) and other public data or private databases. It will also support the common agricultural policy by enabling ML-based solutions applied to image recognition, as well as the use and reuse of data from the Internet of Things (IoT), public data, and user-generated data [148]. FaST relies on

multiple data sources, either connected (online sources) or imported (static sources) into the platform.

## 6. Current Challenges and Future Trends

Given the actual scenario of climate change and economic instability, it is expected that the application of CMs will be boosted because they are able to predict future climate scenarios and test current agricultural systems. Several well-developed models can be found that adapt to different cultivars, environmental conditions, and management practices. A model is just a representation of a real system, and the quality of the results is strongly linked to the quality of the data used by the model. An approach that includes several models at the same time can be quite advantageous in the sense that errors and uncertainties are bridged by the other models. Improving its projection is a major challenge on a spatial scale, although several models can be fused or applied to a given regular grid. Since CMs often present gaps in forecasts, mainly when crop growth conditions deviate from the ideal (biotic and abiotic stresses), data collected by multiple sources of information, namely, remote sensing, can provide information about the actual growth conditions.

The integration of remote sensing data in CMs is, indeed, feasible. Attention must be paid to the correct collection of data, calibration of the sensors, and the correct filling in of the CMs, ensuring that all data are representative of the reality surrounding the crop, and, more specifically, to the field in question. With the increase in knowledge regarding remote sensing techniques and their applicability, as well as the continuous facilitation of accessibility to sensors, the interest in their use in real farm fields has increased, as these sensors are capable of making a quick and expeditious diagnosis of the crop. It is with this advance that the need for boosting adapted sensors arises, and satellite missions are being launched with the capacity to better group radiation into bands and retain a very significant resolution. Applying remote sensing techniques in farmer fields and not just in small experimental plots will contribute to ensuring that they are reliable on a large scale. It will also allow for precision N management, aiming to provide the correct fertilization rate for satisfying the site-specific requirements of crops, both in space and time.

## 7. Conclusions

Nowadays, it is possible to measure and estimate crop N status using rapid, non-destructive measurements. There is a significant relationship between the VIs and the PNC that allows for indirect estimation of the NNI. However, it should be noted that the collection of samples to directly estimate the PNC and the PDM allows for the testing and validation of indirect estimation models, increasing their precision.

There is a wide range of sensors, and the main difference between them is in their practical applicability. Some sensors (leaf-based) accurately estimate the crop N status with a high precision level; however, they cover a very small area, and may even focus only on the leaf level. As discussed throughout this article, the PNC can vary between different parts of the plant, and the leaf N content can provide a different estimate of crop nutrition than the actual PNC. Other sensors (ground-level, UAVs, and satellites) that cover more parts of the plant, or are even able to measure the canopy nutritional status, can provide nutritional data on the crop with great precision, requiring less time to collect information for a much larger area. According to the literature which we reviewed, between these sensor types, UAVs allow imagery with high spatial resolution (centimetric), relatively low operating cost, and near real-time image acquisition to be collected, providing a solution to the problems faced by ground canopy-based and leaf-based sensors. RGB cameras collect high-resolution imagery that is very useful for removing soil background interference in order to estimate the N nutritional status in the vegetative phase of the early stages of the crops' growth. As identified in this review, this is a challenge for Mediterranean crops. Spectral cameras allow for the estimation of indirect parameters related to NNI, such as leaf N content, leaf area index, leaf chlorophyll content, and plant biomass. Finally, existing satellite platforms allow free image acquisition and, consequently, estimate crop N status

in large-scale plots. Attention should be paid to the low resolution of the images, and use should be restricted in the scenario of adverse atmospheric phenomena. Although some knowledge about the determination of PNC via remote sensing is still necessary, mainly in farmer fields, we can define three guidelines to aid in the choice of a correct platform according to objective:

- For measuring the entire field quickly and for free, satellite is the best source of information;
- For quickly measuring the entire field with high resolution and a high level of detail, UAVs can achieve this crop status N estimate;
- For measuring a specific point of the field, leaf-based or ground-level canopy sensors are very accurate in measuring crop N status, and are very adequate for some measurements in the field intended to achieve the real PNC at these specific points.

We highlighted the NIR wavelength bands and the VIs MSAVI2, NDRE, and MCARI to estimate the leaf chlorophyll concentration, NNI, and PNC, respectively. The EVI2 was used for the PDM in areas typically characterized by a Mediterranean climate.

After the diagnosis of crop N status, it is very important to determine how much N fertilizer is needed. In this sense, remote monitoring can be used to adjust models that estimate crop needs. The combination of soil information with the diagnosis of plant N status by spectral measurement could bring about the greatest economic, agronomic, and environmental benefits of the variable rate application of N fertilizers. In Mediterranean agricultural areas, there is great spatial variability in the physical and chemical properties of the soil, even in the smallest fields. Better integration of several types of data, sensors, and algorithms could be carried out to validate field data and to aid in the large-scale application of algorithms. The integration of remote sensing data into crop growth models is feasible through ML models. First, the remote sensing data must be collected by the most suitable sensor, capturing the desired radiation to estimate the desired crop parameter. As for the growth models utilized, they must consider the characteristics of the surrounding crop environment and the management practices that are representative of the region in which they are applied, as well as the parameters of the crop itself.

**Author Contributions:** Conceptualization, L.S.; writing—original draft preparation, L.S.; writing—review and editing, L.S., L.A.C., F.C.L. and B.M. All authors have read and agreed to the published version of the manuscript.

**Funding:** The APC was funded by national funds from Fundação para a Ciência e a Tecnologia (FCT), Portugal, through the research unit UIDP/04035/2020 (GeoBioTec) and by national funds through the Fundação para a Ciência e a Tecnologia, I.P. (Portuguese Foundation for Science and Technology) by the project UIDB/05064/2020 (VALORIZA—Research Centre for Endogenous Resource Valorization).

**Institutional Review Board Statement:** Not applicable.

**Data Availability Statement:** Data sharing is not applicable.

**Acknowledgments:** The authors thank the research centers GeoBioTec and VALORIZA.

**Conflicts of Interest:** The authors declare no conflict of interest.

**Nomenclature**

| | |
|---|---|
| %Ndff | % Nitrogen derived from the fertilizer |
| %Ndfs | % Nitrogen derived from the soil |
| B | Blue |
| CCCI | Canopy Chlorophyll Content Index |
| CHIME | Copernicus Hyperspectral Imaging Mission |
| CM | Crop Model |
| CNI | Canopy Nitrogen Index |
| EC | European Commission |
| ESA | European Space Agency |
| EVI2 | Enhanced Vegetation Index 2 |
| FaST | Farm Sustainability Tool for Nutrients |
| FW | Plant fresh weight |
| IAEA | International Agency Energy Atomic |
| K | Potassium |
| LAI | Leaf area index |
| LSTM | Land Surface Temperature Mission |
| MCARI | Modified Chlorophyll Absorption Ratio Index |
| ML | Machine Learning |
| MM | Multi-Modal |
| MSAVI2 | Modified soil-adjusted vegetation index 2 |
| MTVI2 | Modified Triangular Vegetation Index 2 |
| N | Nitrogen |
| $N_a$ | Real plant nitrogen content |
| $N_c$ | Nitrogen concentration |
| NDRE | Normalized Difference Red-Edge Index |
| NDVI | Normalized Difference Vegetation Index |
| NIR | Near-Infrared |
| NNI | Nitrogen Nutrition Index |
| NNIRS | Nitrogen Nutrition Index Remote Sensing Index |
| NSI | Nitrogen Sufficiency Index |
| NUE | Nitrogen Use Efficiency |
| P | Phosphorus |
| PDM | Plant Dry Matter |
| PNC | Plant Nitrogen Content |
| R | Red |
| RE | Red-Edge |
| RF | Random Forest |
| RVI | Ratio Vegetation Index |
| SDW | Subsample fresh weight |
| SFW | Subsample dry weight |
| SWIR | Short-wave infrared |
| UAV | Unmanned Aerial Vehicle |
| VI | Vegetation Index |
| VIS | Visible bands |
| VRA | Variable rate application |

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
