# Peer review of "Remote Monitoring of Crop Nitrogen Nutrition to Adjust Crop Models: A Review"

_agriculture, doi:10.3390/agriculture13040835_

Round 1
Reviewer 1 Report
Manuscript agriculture-2247057 “Remote monitoring of crop nitrogen nutrition to adjust crop models” is an effort to review crop nitrogen nutrition monitoring through Remote Sensing methods for improvement of crop models.
General Comments
1. Incorporate information about drones in estimation of nitrogen nutrition, although UAV discussed in the article.
2. What are key limitations of crop growth models regarding Nitrogen Use Efficiencies
3. Why certain terms are capitalized? E.g., Vulnerable Areas
4. Define abbreviation at first appearance and subsequently use abbreviated form e.g., NIA, UAV, SWIR, CCCI, CCCI–CNI. If possible add list of abbreviation used.
5. Uniformly correct references, journal names are written inconsistently.
Specific Comments
L-27-28: statement can be improved.
L-40: “…injudicious use of fertilizers without any recommendation…” without any recommendation is confusing, need revision
L-42-45: There are other management practices, proper use of crop rotation by addition of legume crops, slow release fertilizer, precision agriculture techniques
L-47: briefly describe role/factors related to heavy metals
L-48: what type of pollution? Be specific, Nitrogen leaching to ground water or else?
L-48: why capitalize e Fertilizer Use Efficiency?
Instead of “growth states” use growth stages
L-165: Replace “Plants C3” or Plant C4 with “C3 Plants”, C4 Plants
Briefly add reason/s of difference between C3 and C4 plants differ in N dilution curve and N estimation.
L-187: “Read this index requires the…” revise statement
L-234: briefly describe causes of these differences
L-261: rephrase as “efficiency of such sensors is influenced by…”
L-355,356: Revise the text “ Certain satellites have been consid-ered more adapt for regional scale agricultural studies”
Suggested Citations
Ata-Ul-Karim, S. T., Q. Cao, Y. Zhu, L. Tang, M. I. A. Rehmani and W. Cao, 2016: Non-destructive Assessment of Plant Nitrogen Parameters Using Leaf Chlorophyll Measurements in Rice. Frontiers in Plant Science 7, 1829.
Ata-Ul-Karim, S. T., Y. Zhu, Q. Cao, M. I. A. Rehmani, W. Cao and L. Tang, 2017: In-season assessment of grain protein and amylose content in rice using critical nitrogen dilution curve. European Journal of Agronomy 90, 139-151.
Stavrakoudis, D., D. Katsantonis, K. Kadoglidou, A. Kalaitzidis and I. Z. Gitas, 2019: Estimating Rice Agronomic Traits Using Drone-Collected Multispectral Imagery. Remote Sensing 11, 545.
Olson, D. and J. Anderson, 2021: Review on unmanned aerial vehicles, remote sensors, imagery processing, and their applications in agriculture. Agronomy Journal 113, 971-992.
Wang, Y.-P., Y.-C. Chang and Y. Shen, 2022: Estimation of nitrogen status of paddy rice at vegetative phase using unmanned aerial vehicle based multispectral imagery. Precision Agriculture 23, 1-17.
Caturegli, L., M. Corniglia, M. Gaetani, N. Grossi, S. Magni, M. Migliazzi, L. Angelini, M. Mazzoncini, N. Silvestri, M. Fontanelli, M. Raffaelli, A. Peruzzi and M. Volterrani, 2016: Unmanned Aerial Vehicle to Estimate Nitrogen Status of Turfgrasses. PLOS ONE 11, e0158268.
Author Response
Mrs. Luís Silva
15th March 2023
Dear Reviewer,
We would like to thank you for the review effort and dedicated time to evaluate our manuscript. Thanks for your suggestions and corrections.
Please find attached all modifications that were made in accordance with your consideration and the responses to your general and specific comments. In our opinion, significant improvements were carried out across the manuscript.
All the document has been revised to improve the English and make it more correct and readable.
Thanks for your attention.
Yours sincerely,
Mrs. Luís Silva

Reviewer 2 Report
The general approach to this manuscript should be changed. It should be presented as a comprehensive literature review rather than the current form. The current form is misleading. It appears as if the authors have developed a remote monitoring approach for crop nitrogen, which is useful to adjust crop models. In reality, they just recommend such an approach but with no experiments. The manuscript is reviewing approaches for estimating Nitrogen and highlighting their shortfalls. Then, they proposed an approach of combining crop models with remote sensing, but they did not experiment with nor validate that approach. They propose that such an approach will solve the identified problematic issues without even developing such a method.
Abstract:
Although clearly written, the abstract has technical challenges. The challenge is that the method used in this work has not been presented. It is not clear what steps were used to select the literature they reviewed. Was it an arbitrary process? The take-home message is conjecture. They did not develop the method and can only speculate what the method would do.
Introduction
The introduction clearly identified the gap in knowledge and provided a proper motivation to investigate it. Research questions were properly identified. The motivation for studying nitrogen is also clearly and properly motivated.
Method steps lack details. This work suddenly turns into some form of literature review. It is not clear what was done during this analysis. How the literature articles were selected.
The authors need to expand the information on the linkages between vegetation indices from remotely sensed data links to nitrogen availability in plants. All they present is an overview of their conclusions of what each method does and its purported shortfall without thoroughly testing/comparing these methods. That is okay for a literature review manuscript but not with the current approach. They speculate on suitable vegetation indices for this job but do not really test or investigate them.
They need to take out figure1. Its relevance to this topic on nitrogen is very trivial.
The main weakness of the manuscript is the aim. The introduction speaks to techniques of estimating nitrogen BUT the aim is now on "designing methods for approaching and using soil and crop monitoring technologies, and the application of variable dose factors that guarantee productivity and sustainability of crops in Mediterranean regions".
They do not go on to design the methods nor even motivate why the Mediterranean region. The objectives proposed in section 4.2 are not aligned with the knowledge gaps identified as well as motivated in the introduction ["estimation of Nitrogen"]. They should expand the section on the links between crop modeling and the estimation of available nitrogen in plants.
They must design and test their proposed approach to improve this manuscript. Otherwise, make it a literature review
Author Response
Mrs. Luís Silva
15th March 2023
Dear Reviewer,
We would like to thank you for the review effort and dedicated time to evaluate our manuscript. Thanks for your suggestions and comments.
All modifications were made in accordance with your consideration and the answers to your comments and suggestions are in the attached PDF document. In our opinion, significant modifications were carried out across the manuscript allowing us to improve considerably our work.
All the document has been revised to improve the English and make it more correct and readable.
Thanks for your attention.
Yours sincerely,
Mrs. Luís Silva

Round 2
Reviewer 2 Report
Some improvements are noted. However, the manuscript still needs to mature before it can be published. The following points need further attention:
1) The abstract and the conclusion sections of the manuscript have no concluding message. What information should readers get after reading the article? For example, the conclusion from this literature review could be crafted around the direction this type of research is taking or the best-proposed methods for assessing / for doing diagnostics of N status in a non-destructive manner.
2) Missing from this manuscript are key points as follows:
a) The methodological approach for selecting the articles which were analyzed was not provided. The number of articles and the criteria for selecting these articles and the databases from which the articles came should be provided
b) The objectives and/or research questions to guide the development of the manuscript as well as guiding the development of the discussion section were not provided.
Minor comments:
"Should be an emphasis on improving the management of inputs through
measures such as the application of the accurate dose of nitrogen (N) and the application of site-specific fertilizers” ---This is not good grammar or the correct way of starting a sentence
From lines 40 up to line 67, the manuscript discusses the problems with the use of organic fertilizers, heavy metals, and pollution. This manuscript should be focusing on the problems around methods (destructive vs. non-destructive) for determining N status. Whilst a bit of the current context is necessary to provide context, it should be very concise (short). The bigger part should be around methodological gaps in determining the status of N.
Lines 108 to 112. Do not pronounce the research gaps and the subsequent research questions comprehensively. We need to know the gaps in knowledge or science, which will be addressed in this manuscript. We also need to know the research questions, which will guide the development of this paper.
Line 108 “In this paper we’ll address some issues related to assessing crop N nutrition, namely by non-destructive methods, using sensors such as leaf-based, ground-canopy, Unmanned Aerial Vehicles (UAVs) sensors, and satellite platforms”. Be precise-this statement is vague. Provide the identity of the “issues”.
Author Response
Mrs. Luís Silva
20th March 2023
Dear Reviewer,
All modifications were made in accordance to your comments and suggestions. Please find the responses attached. We would like to thanks for the review effort and dedicated time to evaluate our manuscript. In our opinion, significant modifications were carried out across the manuscript allowing us to improve considerably our work.
All the document has been revised to improve the English and make it more correct and readable.
Thanks for your attention.
Yours sincerely,
Mrs. Luís Silva
